# Effect of *Epichloë* Endophyte on the Growth and Carbon Allocation of Its Host Plant *Stipa purpurea* under Hemiparasitic Root Stress

**DOI:** 10.3390/microorganisms11112761

**Published:** 2023-11-13

**Authors:** Peng Zhang, Siyu Meng, Gensheng Bao, Yuan Li, Xiaoyun Feng, Hainian Lu, Jingjuan Ma, Xiaoxing Wei, Wenhui Liu

**Affiliations:** 1Qinghai University, Xining 810003, China; zpkyw1314@163.com (P.Z.); m13591012497@163.com (S.M.); liy19991226@163.com (Y.L.); fengxiaoyun0703@126.com (X.F.); 18794555222@163.com (H.L.); 15294383657@139.com (J.M.); wuiko@163.com (X.W.); qhliuwenhui@163.com (W.L.); 2State Key Laboratory of Sanjiangyuan Ecology and Plateau Agriculture and Animal Husbandry, Qinghai University, Xining 810003, China; 3Qinghai Academy of Animal and Veterinary Medicine, Xining 810016, China

**Keywords:** *Epichloë* endophytes, *Stipa purpurea*, *Pedicularis kansuensis*, hemiparasitic, isotope tracing, carbon

## Abstract

*Epichloë* endophytes not only affect the growth and resistance of their host plants but also confer nutrient benefits to parasitized hosts. In this study, we used *Pedicularis kansuensis* to parasitize *Stipa purpurea*, both with and without endophytic fungi, and to establish a parasitic system. In this study, endophytic fungal infection was found to increase the dry weight of the leaf, stem, and leaf sheath, as well as the plant height, root length, tiller number, aboveground biomass, and underground biomass of *S. purpurea* under root hemiparasitic stress. Meanwhile, the ^13^C allocation of the leaf sheaths and roots of *S. purpurea* increased as the density of *P. kansuensis* increased, while the ^13^C allocation of the leaf sheaths and roots of E+ *S. purpurea* was lower than that of E− *S. purpurea*. The ^13^C allocation of the stem, leaf sheath, and root of E+ *S. purpurea* was higher than that of its E− counterpart. Furthermore, the content of photosynthetic ^13^C and the ^13^C partition rate of the stems, leaves, roots, and entire plant of *S. purpurea* and *P. kansuensis* transferred from *S. purpurea* increased as the density of *P. kansuensis* increased. These results will generate new insights into the potential role of symbiotic microorganisms in regulating the interaction between root hemiparasites and their hosts.

## 1. Introduction

*Epichloë* endophytes commonly grow in the apoplast of the aerial tissues of a variety of cool-season grasses without causing disease [1,2]. Following an extended period of adaptive evolution, a mutually beneficial symbiotic relationship is established between endophytic fungi and cold season grasses. In this relationship, endophytes are capable of acquiring nutrients from their host plants and utilizing the host plant as their habitat sites [3]. Endophytic fungi have been found to influence plant productivity, plant–plant interactions, the structure and biodiversity of plant communities, and the conservation and restoration of ecosystems [4,5]. Thus far, the importance of endophytic fungal communities in mitigating key abiotic stresses has been well studied. However, there are few reports on whether grasses and their symbiotic microorganisms can offset the harmful effects of root hemiparasitic plants in the broader context of biological stress.

Root hemiparasitic plants depend largely on haustoria to obtain nutrients and water from their host plants [6,7] and subsequently weaken their growth [8,9]. Hemiparasitic plants compete with host plants for light resources, which in turn alters the competitive relationship between host and non-host plants. This competition has a significant impact on the composition and structure of plant communities [10,11]. At present, the majority of root hemiparasitic plants have become obnoxious forbs that seriously harm agriculture and animal husbandry [12,13]. For example, *Striga hermonthica* is parasitic on sorghum [14], maize [15], and other crops, resulting in an estimated yield reduction of 65% or even no harvest in some cases. In China, there are many kinds of root hemiparasitic plants, but most are scattered and rarely clustered [16], which has resulted in insufficient attention being paid to the hazardous risks of this group. However, in recent years, some root hemiparasitic plant species have shown significant spreading trends, which pose a potential threat to the sustainable use of grasslands and the healthy development of animal husbandry [17,18]. A typical example is the expanded population of *P. kansuensis* into Qinghai, Xinjiang, southwestern Gansu, western Sichuan, and western Tibet [19].

*P. kansuensis* is an endemic annual or biennial root hemiparasitic herb found in China [20]. Since 2000, the native species of *P. kansuensis* has been expanding rapidly, which has threatened the local livestock industry [17,21]. The hemiparasitic nature of its roots is an exclusive survival strategy employed by *P. kansuensis*. Specifically, it relies on its specialized structure called haustoria to obtain a portion of water and nutrients from gramineous and leguminous plants [22], resulting in the biomass of gramineous and leguminous plants rapidly decreasing [20,23,24,25]. Recent studies have demonstrated that plant symbiotic microorganisms have the ability to greatly enhance their host’s tolerance to root parasitic stress [26,27]. In a study conducted by Sui et al. [28], it was discovered that the growth performance of both hosts and parasites in *Trifolium repens* and *Pedicularis rex/tricolor* legume pairs was greatly enhanced through inoculation with *Glomus mosseae*. This inoculation also helped alleviate the harm caused by *P. rex/tricolor*. At present, most studies focus on *Epichloë* endophytes in improving host grass resistance ability [29]. However, there are currently few reports on the regulation of carbon uptake and allocation by endophytic fungi in grasses and root hemiparasitic systems.

Our previous study found that *P. kansuensis* could establish root parasitic relationships with 18 sympatric plants and that *S. purpurea* was revealed to be the main host of *P. kansuensis* [30]. Meanwhile, the infection rate of endophytic fungi in the degraded grassland of *S. purpurea* was found to be more than 90%. The type of endophytic fungi responsible for the infection was identified as *Epichloë inebrians* [31]. Therefore, in this study, we utilized endophyte-infected (E+) and endophyte-free (E−) *S. purpurea* as our materials to study the plant growth of *S. purpurea* under different intensities of *P. kansuensis* parasitization and normal growth conditions. Accordingly, this study aims to address the following questions: (1) What are the growth characteristics of E+ and E− *S. purpurea* and *P. kansuensis*? (2) What are the total photosynthetic carbon and distribution characteristics of *S. purpurea* when transferred to *P. kansuensis*? (3) How does photosynthetic carbon sequestration and allocation differ between E+ and E− *S. purpurea*?

## 2. Materials and Methods

### 2.1. Experimental Materials

In October 2019, in the natural grassland of Ganzihe Town, Haiyan County, Haibei Prefecture, Qinghai Province (37°07′09″ N, 100°38′42″ E), the mature seeds of *S. purpurea* and *P. kansuensis* were harvested. The harvest site is 3370 m above sea level, which belongs to the classic alpine grassland, the grassland group species is *S. purpurea*, and the associated species are *Agropyron cristatum* and *Poa pratensis*. In our previous study [31,32], we identified the tested material, so we are very sure that it is the seeds of *S. purpurea* and *P. kansuensis*. According to the detection method of endophytic fungi in grass by Li et al. [33], the endophyte infection status of *S. purpurea* was determined by the microscopic examination of leaf sheath pieces and seeds stained with aniline blue [34]; the endophytic fungal carrier rate of *S. purpurea* was detected and found to be as high as 90%. At the same time, 20 seeds of *S. purpurea* were selected for seed skin disinfection and then placed on *Solanum tuberosum* glucose agar medium for 28 days at room temperature and under dark conditions. Referring to the identification method of endophytic fungi in *S. purpurea*, Bao et al. [31] amplified the endophytic fungal sequences, and phylogenetic trees were constructed using Tub, Tef, and Actin specific primers. The endophytic fungus infected with *S. purpurea* was identified as *Epichloë inebrians*. *S. purpurea* E+ seeds were subjected to a soaking process in 70% Topsin-M for eight hours, followed by washing with distilled water to remove any remaining fungicide and to obtain E− seeds.

The experiment was conducted in December 2021 at the smart greenhouse of the Academy of Animal Science and Veterinary Medicine at Qinghai University. E+ and E− seeds of a uniform and complete size were selected and disinfected with 1% sodium hypochlorite for 10 min. The seeds were then rinsed six times with deionized water and blotted with sterilized filter paper to remove any remaining water on the seed surface. Subsequently, the seeds were sown into polyethylene plastic pots measuring 10 cm in diameter and 15 cm in height. To eliminate the influence of nutrients and microorganisms, sand with a diameter of 1–2 mm was used as the growth substrate. The sand had undergone three rounds of rinsing in deionized water and was autoclaved at 120 °C twice for two hours each. After a period of six weeks, the endophytic fungal status of the E+ seedlings was determined to be 100%, while that of the E– seedlings was 0%, according to the method described by Bao et al. [31]. As shown in Figure 1a,b, the arrows refer to the hyphae of *E. inebrians*, and Figure 1c shows that E+ seeds were subjected to a soaking process in 70% Topsin-M for eight hours, followed by washing with distilled water to remove any remaining fungicide and to obtain E− seeds, so its seedings did not have hyphae.

### 2.2. Establishing a Parasitic System

The seed disinfection of *P. kansuensis* was carried out using a method similar to that described above for *S. purpurea*. After disinfection and blotting on sterilized filter paper, six disinfected *P. kansuensis* seeds were sown at a depth of 2 cm into both E+ and E– *S. purpurea* seedlings. The parasitization of *P. kansuensis* onto *S. purpurea* roots was confirmed by the presence of heterogeneous growth of *P. kansuensis* [35]. The experiment was divided into eight treatments based on the endophytic fungal infection status of *S. purpurea* and the parasite density of *P. kansuensis*: (1) *S.purpurea* (E+), (2) *S. purpurea* (E+)+1 *P. kansuensis* seedling, (3) *S. purpurea* (E+)+3 *P. kansuensis* seedling, (4) *S.purpurea* (E−), (5) *S. purpurea* (E−)+1 *P. kansuensis* seedling, (6) *S. purpurea* (E−)+3 *P. kansuensis* seedlings, (7) 1 *P. kansuensis* seedling, (8) 3 *P. kansuensis* seedlings. A fully randomized group arrangement was used, with a total of 128 pots and 16 replications for each of the eight treatments.

### 2.3. Experimental Design and ^13^C Labeling

Within three months of establishing the parasitic relationship between *P. kansuensis* and the E+ and E− type *S. purpurea* plants, a total of six pots were randomly selected from eight treatments. Three pots from each treatment were then placed in a sealed ^13^CO_2_ gas isotope labeling dish with a 60 L internal volume to conduct labeling. The remaining three pots, which were labeled with ^13^C isotopes, were used as the control group. A CO_2_ temperature and gas monitor (AZ7752) was positioned inside the isotope labeling chamber to monitor the concentration of CO_2_ in real time. Additionally, a small fan (N15, 2.5 W) was installed within the chamber to facilitate the mixing of the air and the ^13^CO_2_ gas. To ensure a stable temperature, the outer compartment of the marker box was filled with running water, which ensured that the temperature of the marker box remained between 24 °C and 32 °C. A rubber stopper was attached to the inlet of the isotope-labeling chamber. The labeling gas was obtained from the ^13^CO_2_ bag using a 50 mL syringe and injected into the labeling chamber (Wuhan Estop Technology Co., Ltd., Wuhan, China, purity ≥ 99.8%, abundance ≥ 99%) at a slow pace. Figure 2 shows a scheme of the isotopic labeling box.

The ^13^CO_2_ labeling assay was initiated on 7 July 2022, and labeling was performed daily from 09:00 to 15:00 for three consecutive days. After 3 days of labelling, the samples were taken. The ^13^C-labeled plant was placed in the labeling chamber. The labeling gas from the ^13^CO_2_ canister was introduced once the CO_2_ concentration in the canister dropped to 200 ppm. After introducing 35 mL of ^13^CO_2_ gas, the CO_2_ concentration inside the box was monitored until it dropped to 250–300 ppm. At this point, it was essential to reintroduce another 35 mL of ^13^CO_2_ gas to maintain a stable CO_2_ concentration between 400 and 500 ppm inside the marker box. Additionally, after the third introduction of 35 mL of ^13^CO_2_ gas, regular CO_2_ gas could be introduced to maintain consistent levels of total ^13^CO_2_. To ensure that plant photosynthesis was not affected by the CO_2_ concentration, a No. 8 self-sealing bag (18 × 25 cm) was utilized to cover the *P. kansuensis* in the *S. purpurea*-*P. kansuensis* root parasitic system. Additionally, tin foil was wrapped around the outside of the bag to prevent photosynthesis by *P. kansuensis*, thus eliminating its potential impact on the assay results. Samples of *S. purpurea* and *P. kansuensis* were collected from the treatment and control groups less than one hour after the completion of isotope labeling. The samples were collected from the root, stem, leaf, and leaf sheath of *S. purpurea* and from the root, stem, and leaf of *P. kansuensis*.

### 2.4. Plant Harvest and Analysis

After undergoing multiple washes with deionized water, the samples were subjected to a temperature of 105 °C for a duration of 30 min and subsequently dried in an oven at 65 °C until they reached a constant weight. The samples were then ground and passed through an 80-mesh sieve. The total carbon content and ^13^C content of the collected samples were analyzed using a Vibratome grinding machine (GT200, Beijing Grademan Instruments Co., Ltd., Beijing, China).

The total carbon content and ^13^C abundance were measured using a SerCon Integra 2 (Suzhou Elam Analytical Instruments Co., LTD., Suzhou, China) fully automated stable isotope mass spectrometry system. Both the treatment and control samples were weighed to 0.3 mg (with an accuracy of 0.01 mg) and then packed into tin boats or tin capsules. After placing the sample in the inlet tray, it was combusted in a combustion tube containing silver wire, copper oxide, and chromium oxide at a temperature of 1000 °C. To reduce the amount of gas obtained, a reduction tube filled with copper wire and heated to 600 °C was used. The sample was separated into a column at a temperature of 60 °C after removing the water. An elemental analyzer detector was then used to calculate its carbon content by comparing it to a standard sample. The concentrations and abundances of carbon isotopes were directly measured using a stable isotope mass spectrometry system [36].

### 2.5. ^13^C Calculations

The total carbon content of each organ in the sample, C_i_ (mg) was calculated by multiplying the carbon mass fraction C_i_ (%) of each organ with the biomass C_b_ (g) of that fraction [37]: (1)$$C_{i}\left(mg\right)=C_{i}\left(\%\right)\times C_{b}\left(g\right)\times 10$$

The ^13^C atomic excess percentage of each organ atom%^13^C_i_ excess (i.e., the ^13^C abundance of each organ in the treated group samples (atom^13^C_i_%) minus the background value of ^13^C abundance of each organ in the control group samples (atom^13^C_o_%)) was calculated as follows: (2)$${atom}^{13}C_{i}\;\%\;excess={atom}^{13}C_{i}\%-{atom}^{13}C_{o}\%$$

To determine the amount of ^13^C photosynthesis in each organ in the sample, we used Equation (3) by substituting the ^13^C atomic percentage. The variable ^13^C_i_ represents the amount of ^13^C photosynthesis in each organ in the sample (mg), and C_i_ represents the quantity of carbon in each organ (mg): (3)$${}^{13}C_{i}\left(mg\right){=C}_{i}\left(mg\right)\times atom{}^{13}C_{i}\%\;excess/100$$

The fractional ^13^C allocation rate Pi% could be calculated using Equation (4), where ^13^C_i_ represents the amount of ^13^C photosynthesis in each organ in the sample (mg), and Σ^13^C_i_ represents the total amount of ^13^C photosynthesis in each organ of the plant sampled (mg): (4)P1i%=13Ci∑13Ci×100

### 2.6. Data Analysis

All statistical analyses were conducted using IBM SPSS Statistics Version 26.0. Before performing the analyses, the data were tested for normality using the Shapiro–Wilk test and for homogeneity of variance using the Brown–Forsythe test. Statistical analyses for determining significant differences in observations were applied throughout the study using two-way ANOVA, one-way ANOVA, or Student’s *t*-tests. The significance level was set at *p* < 0.05 for all statistical analyses.

## 3. Results

### 3.1. Effects of Endophytic Fungal Infection and P. kansuensis Parasitism on S. purpurea and P. kansuensis Growth Characteristics

Under root hemiparasitic stress, there was a significant effect of endophytic fungal infection on the dry weight of the leaf, stem, leaf sheath, tiller number, and aboveground biomass and underground biomass (Table 1, *p* < 0.01). Meanwhile, the parasitic density of *P. kansuensis* had a significant effect on the dry weight of the leaf, stem, and leaf sheath, as well as the plant height, root length, tiller number, aboveground biomass, and underground biomass of *S. purpurea* (Table 1, *p* < 0.01). Apart from the plant height and tiller number, the interaction between the endophytic fungal infection and *P. kansuensis* parasitic density had a significant effect on the rest of the growth characteristics of *S. purpurea* (Table 1, *p* < 0.01). As the parasitic density of *P. kansuensis* increased, apart from the dry weight of the leaf sheath, the dry weight of the leaf, stem, aboveground biomass, underground biomass, plant height, root length, and tiller number of *S. purpurea* exhibited a downward trend (Figure 3a–h). When *S. purpurea* was parasitized by *P. kansuensis*, the dry weight of the leaf, stem, and leaf sheath, as well as the tiller number, aboveground biomass, and underground biomass of E+ *S. purpurea*, were significantly higher than those of E− plants (Figure 3a–h, *p* < 0.05). Without the parasitic effect of *P. kansuensis* on *S. purpurea*, the dry weight of the leaf and stem, as well as the root length, tiller number, aboveground biomass, and underground biomass, were the highest compared to those of samples receiving other treatments (Figure 3a–h).

The parasitic density had a significant effect on the aboveground and underground biomass of *P. kansuensis* (Table 2, *p* < 0.01). However, it did not have an effect on the plant height, root length, or underground biomass (Table 2, *p* > 0.05). At the same time, the endophytic fungal infection status of *S. purpurea* had significant effects on the root length, as well as the aboveground and underground biomasses of *P. kansuensis* (Table 2, *p* < 0.05). However, it did not have an effect on the plant height (Table 2, *p* > 0.05). In addition, the interaction between endophytic fungal infection and *P. kansuensis* parasitic density had a significant effect on the underground biomass of *P. kansuensis*, but there was no impact on the plant height, root length, and aboveground biomass (Table 2, *p* < 0.01). As the parasitic density of *P. kansuensis* increased, there was a noticeable declining trend in plant height, but there was a noticeable increasing trend in both the aboveground and underground biomasses of *P. kansuensis* (Figure 4a–d). Furthermore, when there were no host plants, the plant height, root length, and aboveground and underground biomasses of *P. kansuensis* increased with the increase in plant density (Figure 4a–d). Additionally, for *P. kansuensis* of parasitized E+ *S. purpurea*, the plant height and root length were lower than those of parasitized E− plants (Figure 4a,b), and the aboveground and underground biomasses were higher than those of parasitized E− plants (Figure 4a,b).

The maximum number of *P. kansuensis* haustoria per unit area in the root of *S. purpurea* was the highest (i.e., 0.36 cm^2^) when there was no endophytic fungal infection and only one parasite of *P. kansuensis* (Figure 5). Conversely, the lowest maximum number of *P. kansuensis* haustoria per unit area in the root of *S. purpurea* was 0.12 cm^2^, which occurred when there was both endophytic fungal infection and one parasite of *P. kansuensis* (Figure 5). As the density of *P. kansuensis* parasitica increased, the haustoria number of *P. kansuensis* parasitica per unit area in the root area of E+ *S. purpurea* increased, while this haustoria number decreased in the root area of E− *S. purpurea*, although the differences were not significant (Figure 5, *p* > 0.05).

### 3.2. Effects of Endophyte Fungal Infection and P. kansuensis Parasitica on the Total Carbon Content of S. purpurea and P. kansuensis

Endophytic fungal infection and *P. kansuensis* parasitica had a significant impact on the total carbon content of the leaf sheath, stem, leaves, and roots of *S. purpurea* under root hemiparasitic stress (Table 3, *p* < 0.01). Additionally, there was a significant interaction between endophytic fungus infection and *P. kansuensis* parasitica density that affected the total carbon content of the leaf sheath, stem, leaves, and roots of *S. purpura* (Table 3, *p* < 0.01). With the increase in the parasitic density of *P. kansuensis*, the total carbon content of the stems, leaves, and roots decreased, but this was not the case with the leaf sheath (Figure 6a–d). Additionally, the total carbon contents of the leaf sheath, stem, leaves, and roots of E+ *S. purpura* were significantly higher than those of the E− plants, regardless of whether the plant was parasitic or not (Figure 6a–d, *p* < 0.05).

The endophytic fungal infection status of *S. purpurea* and the parasitic density of *P. kansuensis* were significantly affected by the total carbon content in the stems, leaves, and roots of *P. kansuensis* (Table 4, *p* < 0.01). Specifically, the total carbon content of the stems, leaves, and roots of *P. kansuensis* was significantly influenced by the interaction between the endophytic fungal infection status of *S. purpurea* and the parasitic density of *P. kansuensis* (Table 4, *p* < 0.05). The total carbon contents of the stems, leaves, and roots of *P. kansuensis* increased as the density of *P. kansuensis* increased, irrespective of the presence of its host *S. purpurea* (Figure 7a,b). At the same time, the total carbon contents of the stems, leaves, and roots of the *P. kansuensis* parasiticum E+ *S. purpurea* were higher than those of the *P. kansuensis* parasiticum E− *S. purpurea* (Figure 7a,b), but the total carbon contents of the stems, leaves, and roots of the *P. kansuensis* parasiticum E− *P. kansuensis* were lower than those of *P. kansuensis* without a host (Figure 7a,b).

### 3.3. Effects of Endophyte Fungal Infection and P. kansuensis Parasitica on the Photosynthetic ^13^C Content of S. purpurea and P. kansuensis

Under root hemiparasitic stress, both endophytic fungal infection and the parasitic density of *P. kansuensis* had significant impacts on the leaf sheath, stem, leaves, roots, and overall photosynthetic ^13^C content (Table 5, *p* < 0.01). Specifically, the interaction between endophytic fungal infection and *P. kansuensis* parasitica had a significant impact on the photosynthetic ^13^C content in the leaf sheath, leaf, stem, and root of *S. purpurea* (Table 5, *p* < 0.05). With the increasing parasitic density of *P. kansuensis*, the photosynthetic ^13^C content in the leaves, stems, and overall structures of E+ and E− *S. purpurea* showed a decreasing trend, while the same content of the leaf sheath and root showed no significant change (Figure 8a–e). The photosynthetic ^13^C contents in the leaf sheath, stem, leaves, and overall structure of E+ *S. purpurea* were significantly higher compared to the corresponding values for E− plants (Figure 8a–e, *p* < 0.05).

The parasitic density and endophytic fungal infection status had significant effects on the ^13^C content of the stems, leaves, and overall photosynthesis transferred to *S. purpurea* and *P. kansuensis* (Table 6, *p* < 0.01), and the parasitic density had significant effects on the ^13^C content of roots photosynthesis transferred to *S. purpurea* and *P. kansuensis* (Table 6, *p* < 0.01). However, there was no effect on the photosynthetic ^13^C content of *P. kansuensis* that was transferred from the roots of *S. purpurea* (Table 6, *p* = 0.61). At the same time, the interaction of the endophytic fungal infection status of *S. purpurea* and the parasitic density of *P. kansuensis* had a significant effect on the photosynthetic ^13^C content of *P. kansuensis* that was transferred from the stems and leaves of *S. purpurea* (Table 6, *p* < 0.01). However, it was observed that the photosynthetic ^13^C content of the roots and overall structure of *P. kansuensis* did not show any significant changes when transferred from the stems and leaves of *S. purpurea* (Table 6, *p* > 0.05). However, as the parasitic density of *P. kansuensis* increased, there was a continuous increase in the ^13^C content of the stem leaves, roots, and overall photosynthesis of *P. kansuensis* transferred from *S. purpurea* (Figure 9a–c). The ^13^C content in the stems, leaves, roots, and overall photosynthetic structure of *P. kansuensis* that was transferred from E+ *S. purpurea* was higher than that of E− *S. purpurea* (Figure 9a–c).

### 3.4. Effects of Endophyte Fungal Infection and P. kansuensis Parasitica on the Photosynthetic ^13^C Allocation Rate of S. purpurea and P. kansuensis

The endophytic fungal infection and the density of *P. kansuensis* parasites had a significant impact on the allocation of photosynthetic ^13^C in the leaves, stems, leaf sheaths, and roots of *S. purpurea* when subjected to root hemiparasitic stress (Table 7, *p* < 0.05). Among these components, the interaction between endophytic fungal infection and the parasitic density of *P. kansuensis* had a significant effect on the photosynthetic ^13^C allocation rate in the leaves, stems, leaf sheath, and roots of *S. purpurea* (Table 7, *p* < 0.01). As the parasitic density of *P. kansuensis* increased, the photosynthetic ^13^C allocation rates of E+ and E− *S. purpurea* in the leaf sheath and roots exhibited an upward trend. However, there was no significant change in the photosynthetic ^13^C content in the leaves and stems (Figure 10a–d). Under the condition of *P. kansuensis* parasitism, the photosynthetic ^13^C allocation rate of E+ *S. purpurea* leaves was lower than that of E− *S. purpurea*. However, the rate of photosynthetic ^13^C allocation in E+ *S. purpurea* stems, leaf sheaths, and roots was higher than that of E− *S. purpurea* (Figure 10a–d). Under the condition of three-parasitic *P. kansuensis*, the photosynthetic ^13^C allocation rates of the leaves, leaf sheath, and roots of E+ *S. purpurea* were higher than those of E− *S. purpurea*, while the photosynthetic ^13^C allocation rate of the stems of E + *S. purpurea* was lower than that of E− *S. purpurea* (Figure 10a–d).

The endophytic fungal infection and the parasitic density of *P. kansuensis* had significant effects on the ^13^C carbon allocation rate of the stem leaves and roots of *P. kansuensis* (Table 8, *p* < 0.01). Among these effects, the endophytic fungal infection status of *S. purpurea* and the parasitic density of *P. kansuensis* had a significant interaction effect on the ^13^C allocation rate of the stem leaves and roots of *P. kansuensis* (Table 8, *p* < 0.01). As the parasitic density of *P. kansuensis* increased, the ^13^C allocation rate of the stem leaves of *P. kansuensis* continued to decrease, and the ^13^C allocation rate of the roots of *P. kansuensis* continued to increase (Figure 11a,b). At the same time, the ^13^C allocation rate of *P. kansuensis* stem leaves parasitic upon E+ *S. purpurea* was higher than that of E− plants (Figure 11a, *p* < 0.05). However, when the parasitic density was one, the ^13^C allocation rate of *P. kansuensis* roots parasitic upon E+ *S. purpurea* was lower than that of E− plants (Figure 11b, *p* < 0.05).

## 4. Discussion

Root parasite plants usually rob their hosts of water, carbohydrates, and nutrients to meet their own growth needs [7,38]. Numerous studies have demonstrated that root parasitic plants can negatively impact the growth characteristics of their host plants [39,40]. In our experiment, we observed that when *P. kansuensis* parasitized *S. purpurea*, there was a decrease in the leaf weight, stem weight, sheath weight, aboveground biomass, underground biomass, plant height, root length, and tillering number of *S. purpurea* (Figure 3a–h). Meanwhile, as the parasitic density of *P. kansuensis* increased, the inhibitory effect also increased gradually. 

Some studies have suggested that endophytic fungi can enhance a plant’s ability to withstand both biotic and abiotic stresses [41,42,43]. In the context of root parasitic stress, our study revealed that E+ *S. purpurea* exhibited a greater leaf weight, stem weight, sheath weight, aboveground biomass, underground biomass, root length, and tillering number compared to E− plants (Figure 3a–h). In conclusion, we found that endophytic fungal infection can enhance the tolerance of *S. purpurea* to the root hemiparasitic stress caused by *P. kansuensis*. Furthermore, when *P. kansuensis* parasitized *S. purpurea* with an endophytic fungal infection, it had varying effects on the plant’s height, root length, aboveground biomass, and underground biomass.

It is worth noting that after *P. kansuensis* parasitized E+ *S. purpurea*, we observed that its stems and leaves easily withered, but new stems and leaves could be regrown. This could be due to the infection of endophytic fungi inhibiting *P. kansuensis* from getting nutrients from *S. purpurea*, thereby causing the living environment of *P. kansuensis* to deteriorate. However, in the case of *P. kansuensis* parasitized with E+ *S. purpurea*, its aboveground and underground biomasses were higher than those of *P. kansuensis* parasitized with E− *S. purpurea* (Figure 4c,d); the reason for this difference may be that the endophytic fungus infection may enhance the photosynthetic capacity of *S. purpurea* under stressful conditions. This can lead to the accumulation of more carbohydrates, which would enable *S. purpurea* and root parasitic plants to meet their growth requirements. Consequently, endophytic fungi help alleviate the root hemiparasitic damage caused by *P. kansuensis* on *S. purpurea*. 

In this study, we found that when the parasitic density of *P. kansuensis* was varied, the aboveground, underground, and total biomasses of the E+ *S. purpurea* and *P. kansuensis* parasitism systems were higher compared to the E− S. purpurea and *P. kansuensis* parasitism systems. The results of this study suggest that endophytic fungi enhance the light capture capacity of the aboveground part of *S. purpurea* by influencing its growth characteristics. Additionally, endophytic fungi promote root growth and improve a plant’s ability to compete for nutrient absorption in the subsurface. By reducing the number of haustoria of *P. kansuensis* per unit area, the nutrient plunder from *S. purpurea* is minimized, resulting in an increased accumulation of dry matter in the host–parasitic plant system.

The process of plant growth is influenced by both material accumulation and physiological metabolism, which are, in turn, affected by plant photosynthesis. Approximately 90% of the materials required for plant growth are derived from organic compounds produced through photosynthesis, which typically have a carbon content ranging from 40 to 50% [44]. Photosynthetic carbon is the starting point of carbon metabolism in plants, which is the most important metabolic activity in the process of plant growth and development. This function mainly includes the assimilation of inorganic carbon (CO_2_) into organic carbon and the metabolic processes of carbohydrate transformation, transport, accumulation, and decomposition among different plant tissues [45,46]. Thus, the quantitative allocation and fixation of photosynthetic carbon is of great significance for studying the allocation of fixed carbon in plants, while also serving as an important means by which to analyze the influence of external factors on plant growth [46,47].

Parasitic plants use the root haustorium to establish a parasitic relationship with the host, and part of the carbohydrate produced by the host through photosynthesis is ingested by parasitic plants through phloem channels [7,48]. Studies have shown that root hemiparasitic plants can directly inhibit the photosynthetic metabolism of hosts and reduce their aboveground biomass [49,50,51,52]. In this study, with the increase in the parasitic density of *P. kansuensis*, the total carbon contents of the leaves, stems, and roots of *S. purpurea* decreased, and the total carbon contents of E+ *S. purpurea* leaves, stems, leaf sheaths, and roots were higher than those of E− plants (Figure 6a–d). This is consistent with Suryanarayanan et al.’s conclusion that endophytic fungi have a positive effect on the photosynthetic characteristics of plants [53]. In addition, with the increase in the density of *P. kansuensis*, the total carbon content of the stem leaves and roots of *P. kansuensis* increased, regardless of whether there was a host, and the total carbon contents of the stem leaves and roots of *P. kansuensis* parasitizing E+ *S. purpurea* were higher than those of *P. kansuensis* parasitica E− plants. However, the total carbon contents of the stem leaves and roots of *P. kansuensis* parasitica E− *S. purpurea* were lower than those of plants without a parasite (Figure 8a,b). On the one hand, the inhibition of photosynthesis in E− *S. purpurea* under root hemiparasitic stress could have led to a decrease in the carbon allocation absorbed by *P. kansuensis*. On the other hand, in the pot experiment, both *S. purpurea* and *P. kansuensis* may have competed for nutrients, resulting in insufficient nutrient availability for both species. In contrast, the host-free *P. kansuensis* could fully absorb the required nutrients.

Up to now, the positive effect of endophytic fungi on photosynthetic carbon fixation has been widely accepted [53], but its mechanism needs to be further explored. ^13^C pulse labeling can be used to study the distribution of photosynthetic carbon in plants at a certain period in time [47,54,55]. Therefore, ^13^C pulse labeling can be used to study the photosynthetic carbon allocation in the parasitic system established by host–parasite plants [37]. In this study, ^13^CO_2_ isotope labeling was used to quantitatively analyze the photosynthetic ^13^C amount and ^13^C allocation rate of *P. kansuensis* that was transferred from *S. purpurea* and the total carbon content of *P. kansuensis*. In our experiment, the stem photosynthetic 13C of high-density parasitic E+ *S. purpurea* was significantly higher compared to that of non-parasitic E+ plants (Figure 8b, *p* < 0.05), while that of low-density parasitic E+ *S. purpurea* was higher than that of non-parasitic E+ *S. purpurea*, although the differences were not significant (Figure 8b, *p* > 0.05). This may be the result of the transition from a mutualistic relationship to a competitive one between endophytic fungi and *S. purpurea*. Some studies have shown that endophytic fungi are most frequently distributed in the stem internode and leaf sheath [56], and the difference in the photosynthetic ^13^C in the stem and leaf sheath of E+ *S. purpurea* under different parasitic densities may be the result of different distribution densities of endophytic fungi. At the same time, compared with the high-density parasitism in *P. kansuensis*, carbon fixed by photosynthesis and carbohydrates formed by carbon cycling can meet the growth requirements of *S. purpurea*, endophytic fungi, and root parasitism plants. However, in our study, the carbon and carbohydrate synthesized from the stems and leaves of *S. purpurea* could not meet the above three requirements at the same time under the high-density parasitic stress of *P. kansuensis*. To maintain its normal growth, *S. purpurea* preferentially delivers more carbon and carbohydrates to the roots, which is consistent with the result that the root biomass of E+ *S. purpurea* was higher than that of E− *S. purpurea* under heavy root parasitic stress. This conclusion is also consistent with the survival strategy that plants preferentially distribute nutrients to the roots, through which they also obtain nutrient resources in the soil, under the condition of nutrient scarcity [57]. 

It Is worth noting that under conditions of heavy parasitism, the ^13^C allocation rates of the stem and root of E+ *S. purpurea* were lower than those of E− *S*. *purpurea* (Figure 10a–d, *p* > 0.05), although the differences were not significant. However, the photosynthetic ^13^C amounts in the stem and root of E+ *S*. *purpurea* were significantly higher compared to those of E− plants (Figure 8a–d, *p* < 0.05). These results indicate that the carbon metabolic efficiency of E+ *S. purpurea* is higher than that of E− plants under the condition of heavy parasitism of *P. kansuensis*, which further supports the idea that endophytic fungi can improve the photosynthetic capacity of host grasses and thus enhance the carbon utilization efficiency. Meanwhile, the photosynthetic ^13^C amounts in the stems, leaves, and whole plants of *P. kansuensis* that were transferred from E+ *S. purpurea* were higher than those transferred from E− *S. purpurea*, but there was no difference in the photosynthetic ^13^C amounts of *P. kansuensis* transferred from the roots (Figure 9a–c). Furthermore, the photosynthetic ^13^C allocation rate of *P. kansuensis* transferred from *S. purpurea* further supports the above view. Compared with *P. kansuensis*, which was parasitic upon E− *S. purpurea*, the photosynthetic allocation rate of ^13^C in the stem leaves of *P. kansuensis* parasitized by E+ *S. purpurea* was higher, but the photosynthetic allocation rate of ^13^C was significantly decreased when the roots were transferred from *S. purpurea* (Figure 11a,b). These results further indicate that endophytic fungi regulate the distribution ratio of photosynthetic ^13^C in the roots and stem leaves of *P. kansuensis* when transferred from *S. purpurea* in an environment of heavy parasitism. In other words, *P. kansuensis*’s stem leaves receive a preferential distribution of photosynthetic ^13^C, which makes the root’s primary source of carbon photosynthetic carbon, which is then used to offset the plant’s excessive uptake of *S. purpurea*’s carbohydrates. At the same time, the regulatory mechanism of photosynthetic carbon allocation by endophytic fungi on the transfer of *P. kansuensis* from *S. purpurea* could further explain why *P. kansuensis* accumulated more biomass when parasitized by E+ *S. purpurea* (Figure 4). 

## 5. Conclusions

Our study shows that parasitism by *P. kansuensis* can inhibit the growth of *S. purpurea,* and the inhibitory effect increases with an increase in the parasitic density of *P. kansuensis*. Compared with the E− *S. purpurea*, the plant height, root length, and tiller number of the infected *S. purpurea* were significantly increased. The results indicate that endophytic fungal infection can increase the total carbon content of *S. purpurea* under root hemiparasitic stress. Meanwhile, the total carbon contents of stems, leaves, leaf sheathes, and roots were significantly inhibited by *P. kansuensis* parasitism, and this effect worsened with the increase in the parasitic density of *P. kansuensis*. However, endophytic fungal infection was able to increase the total carbon content of *S. purpurea* under parasitic stress. Under the heavy parasitism of *P. kansuensis*, the carbon metabolic efficiency of E+ *S. purpurea* was higher than that of E− plants. Finally, endophytic fungi regulate the distribution ratio of photosynthetic ^13^C in the roots and stem leaves of *P. kansuensis* transferred from *S. purpurea*, with a preference for distribution to the stem leaves of *P. kansuensis*. This study focused solely on the growth and carbon distribution of *S. purpurea* under the root parasitism stress caused by endophytic fungi; further research is needed to investigate the allocation of other elements such as nitrogen and phosphorus. 

## Figures and Tables

**Figure 1 microorganisms-11-02761-f001:**
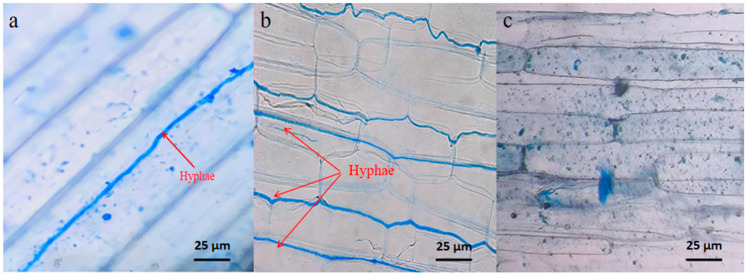
Image of endophytic fungal hyphae of *S. purpurea* leaf sheath. Note: (**a**) is a hyphae image of the microscopic examination of the *S. purpurea* leaf sheath collected from natural grassland; (**b**) is a hyphae image of the microscopic examination of the seedlings’ leaf sheath of endophyte infection *S. purpurea*(E+) used in the experiments; (**c**) is an image of the microscopic examination of the seedlings’ leaf sheath of endophyte none-infection *S. purpurea*(E−) used in the experiments.

**Figure 2 microorganisms-11-02761-f002:**
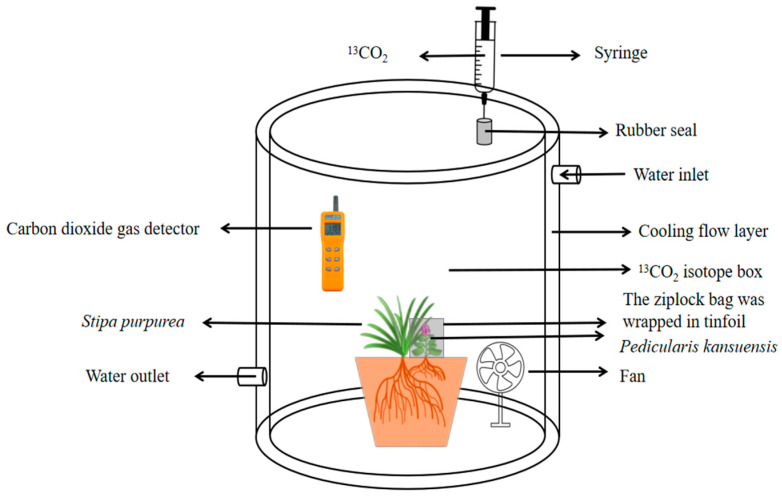
Schematic diagram of ^13^C−CO_2_ isotope labeling for root parasitized systems between *S. purpurea* and *P. kansuensis*.

**Figure 3 microorganisms-11-02761-f003:**
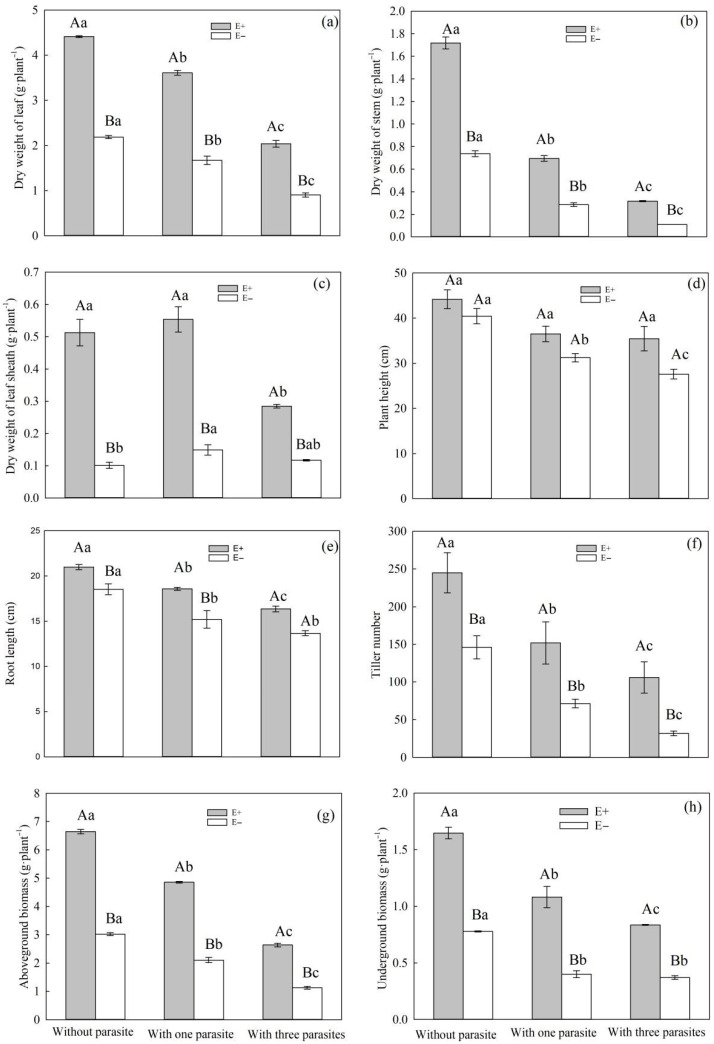
Effects of *P. kansuensis* density and endophyte status on the growth of *S. purpurea*. Note: Data are the mean standard errors. Different capital letters indicate significant differences between different *S. purpurea* endophyte statuses at the same *P. kansuensis* density (*p <* 0.05). Different lowercase letters indicate significant differences between different *P. kansuensis* densities in the same *S. purpurea* endophyte state (*p <* 0.05). One hemiparasite indicates that one *S. purpurea* plant was parasitized by one *P. kansuensis*, three hemiparasites indicate that one *S. purpurea* plant was parasitized by three *P. kansuensis*, and “without parasite” indicates *S. purpurea* growing alone; (**a**) Effects of *P. kansuensis* density and endophyte status on the dry weight of leaf of *S. purpurea*; (**b**) Effects of *P. kansuensis* density and endophyte status on the dry weight of stem of *S. purpurea*; (**c**) Effects of *P. kansuensis* density and endophyte status on the dry weight of leaf sheath of *S. purpurea*; (**d**) Effects of *P. kansuensis* density and endophyte status on the plant height of *S. purpurea*; (**e**) Effects of *P. kansuensis* density and endophyte status on the root length of *S. purpurea*; (**f**) Effects of *P. kansuensis* density and endophyte status on the tiller number of *S. purpurea*; (**g**) Effects of *P. kansuensis* density and endophyte status on the aboveground biomass of *S. purpurea*; (**h**) Effects of *P. kansuensis* density and endophyte status on the underground biomass of *S. purpurea*.

**Figure 4 microorganisms-11-02761-f004:**
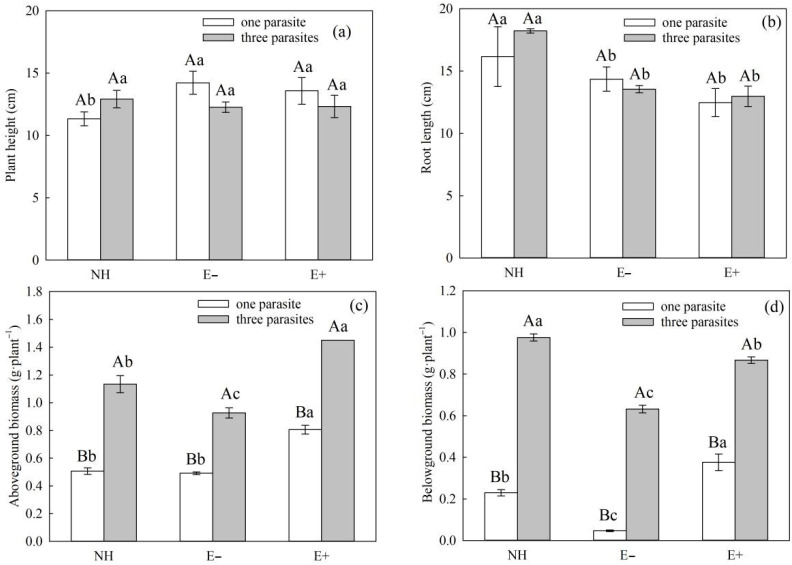
Effects of the *P. kansuensis* density and endophyte status of *S. purpurea* on the growth of *P. kansuensis*. Note: Data are the mean standard errors. Different capital letters indicate significant differences between the different *P. kansuensis* densities in the same *S. purpurea* endophyte state (*p* < 0.05); different lowercase letters indicate significant differences between the different *S. purpurea* endophyte states at the same *P. kansuensis* density (*p* < 0.05). NH indicates that *P. kansuensis* was not grown with *S. purpurea*; (**a**) Effects of the *P. kansuensis* density and endophyte status of *S. purpurea* on the plant height of *P. kansuensis*; (**b**) Effects of the *P. kansuensis* density and endophyte status of *S. purpurea* on the root length of *P. kansuensis*; (**c**) Effects of the *P. kansuensis* density and endophyte status of *S. purpurea* on the aboveground biomass of *P. kansuensis*; (**d**) Effects of the *P. kansuensis* density and endophyte status of *S. purpurea* on the underground biomass of *P. kansuensis*.

**Figure 5 microorganisms-11-02761-f005:**
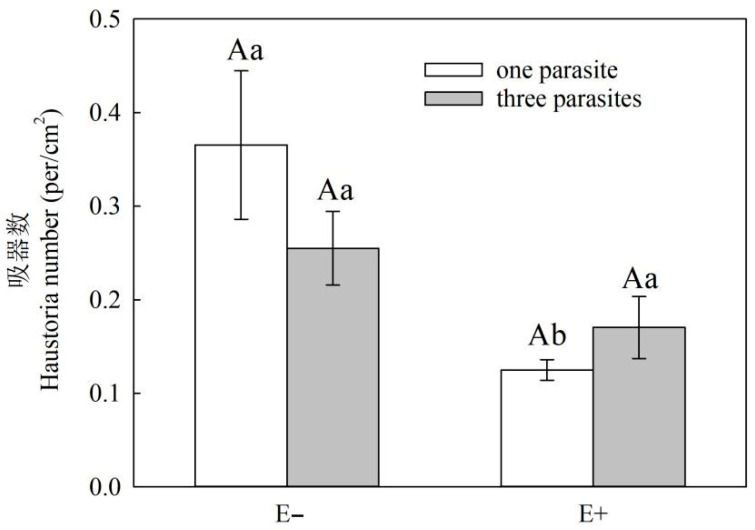
Effect of *S. purpurea* endophyte status on the number of haustoria per unit area of parasitic systems of *S. purpurea* and *P. kansuensis*. Note: The number of *P. kansuensis* haustoria is based on the body microscope (OlyMPUS DP74); a root scanner (model EPSON7500, a resolution of 400 BPI) was used to scan the fresh purple flower needle and the root surface area; the number of *P. kansuensis* haustoria per unit area is the ratio of the number of haustoria of *P. kansuensis* to the basis of the root surface area. Data are the mean standard errors. Different capital letters indicate significant differences between different *S. purpurea* endophyte statuses at the same *P. kansuensis* density (*p* < 0.05). Different lowercase letters indicate significant differences between different *P. kansuensis* densities in the same *S. purpurea* endophyte state (*p* < 0.05).

**Figure 6 microorganisms-11-02761-f006:**
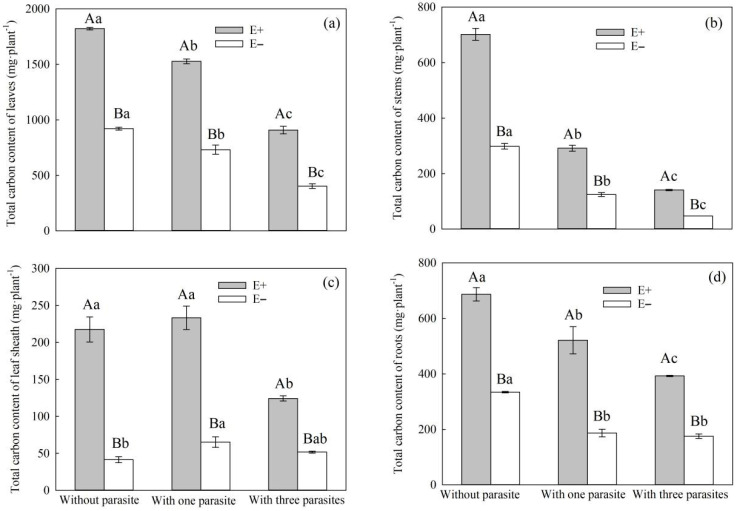
Effects of *P. kansuensis* density and endophyte status on the total carbon content of leaves, stems, leaf sheaths, and roots of *S. purpurea*. Note: Data are the mean standard errors. Different capital letters indicate significant differences between different *S. purpurea* endophyte statuses at the same *P. kansuensis* density (*p* < 0.05). Different lowercase letters indicate significant differences between different *P. kansuensis* densities in the same *S. purpurea* endophyte state (*p* < 0.05). One hemiparasite indicates that one *S. purpurea* plant was parasitized by one *P. kansuensis*, three hemiparasites indicate that one *S. purpurea* plant was parasitized by three *P. kansuensis*, and “without parasite” indicates *S. purpurea* growing alone; (**a**) Effects of *P. kansuensis* density and endophyte status on the total carbon content of leaves of *S. purpurea*; (**b**) Effects of *P. kansuensis* density and endophyte status on the the total carbon content of stems of *S. purpurea*; (**c**) Effects of *P. kansuensis* density and endophyte status on the the total carbon content of leaf sheaths of *S. purpurea*; (**d**) Effects of *P. kansuensis* density and endophyte status on the total carbon content of roots of *S. purpurea*.

**Figure 7 microorganisms-11-02761-f007:**
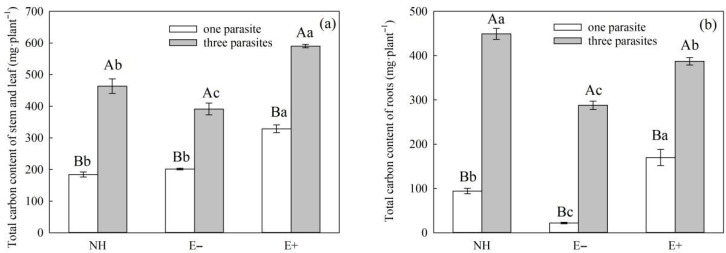
Effects of *P. kansuensis* density and endophyte status on the total carbon content of the stems, leaves, and roots of *P. kansuensis*. Note: Data are the mean standard errors. Different capital letters indicate significant differences between the different *P. kansuensis* densities in the same *S. purpurea* endophyte state (*p* < 0.05). Different lowercase letters indicate significant differences between the different *S. purpurea* endophyte states at the same *P. kansuensis* density (*p* < 0.05). NH indicates that *P. kansuensis* was not grown with *S. purpurea*; (**a**) Effects of *P. kansuensis* density and endophyte status on the total carbon content of leaves of *P. kansuensis*; (**b**) Effects of *P. kansuensis* density and endophyte status on the the total carbon content of stems of *P. kansuensis*.

**Figure 8 microorganisms-11-02761-f008:**
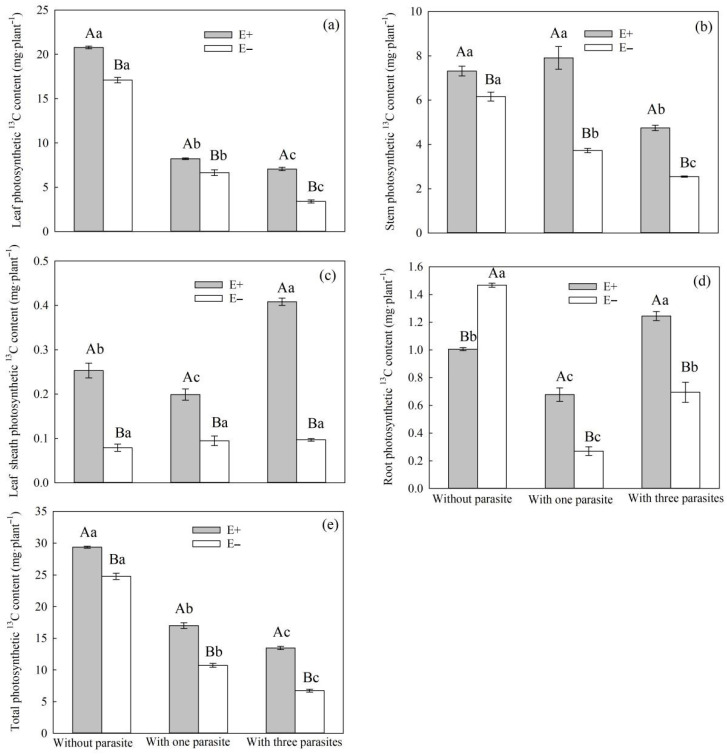
Effects of *P. kansuensis* density and endophyte status on the leaves, stems, sheath, root, and total photosynthetic ^13^C content of *S. purpurea*. Note: Data are the mean standard errors. Different capital letters indicate significant differences between different *S. purpurea* endophyte statuses at the same *P. kansuensis* density (*p* < 0.05). Different lowercase letters indicate significant differences between different *P. kansuensis* densities in the same *S. purpurea* endophyte state (*p* < 0.05). One hemiparasite indicates that one *S. purpurea* plant was parasitized by one *P. kansuensis*, three hemiparasites indicate that one *S. purpurea* plant was parasitized by three *P. kansuensis*, and “without parasite” indicates *S. purpurea* growing alone; (**a**) Effects of *P. kansuensis* density and endophyte status on the leaves photosynthetic ^13^C content of *S. purpurea*; (**b**) Effects of *P. kansuensis* density and endophyte status on the stems photosynthetic ^13^C content of *S. purpurea*; (**c**) Effects of *P. kansuensis* density and endophyte status on the sheath photosynthetic ^13^C content of *S. purpurea*; (**d**) Effects of *P. kansuensis* density and endophyte status on the root photosynthetic ^13^C content of *S. purpurea*; (**e**) Effects of *P. kansuensis* density and endophyte status on the total photosynthetic ^13^C content of *S. purpurea*.

**Figure 9 microorganisms-11-02761-f009:**
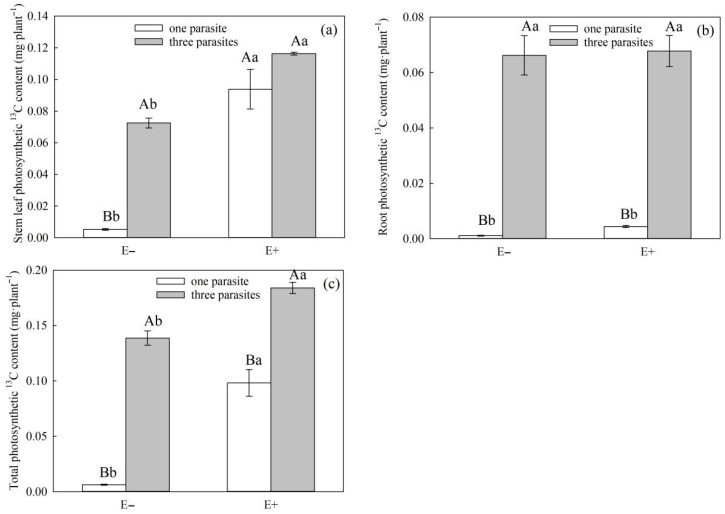
Effects of the *P. kansuensis* density and endophyte status on the ^13^C content of the stem leaves, roots, and total plant of *P. kansuensis*. Note: Data are the mean standard errors. Different capital letters indicate significant differences between different *S. purpurea* endophyte statuses at the same *P. kansuensis* density (*p* < 0.05); Different lowercase letters indicate significant differences between different *P. kansuensis* densities in the same *S. purpurea* endophyte state (*p* < 0.05); (**a**) Effects of *P. kansuensis* density and endophyte status on the stem leaves photosynthetic ^13^C content of *S. purpurea*; (**b**) Effects of *P. kansuensis* density and endophyte status on the roots photosynthetic ^13^C content of *S. purpurea*; (**c**) Effects of *P. kansuensis* density and endophyte status on the total photosynthetic ^13^C content of *S. purpurea*.

**Figure 10 microorganisms-11-02761-f010:**
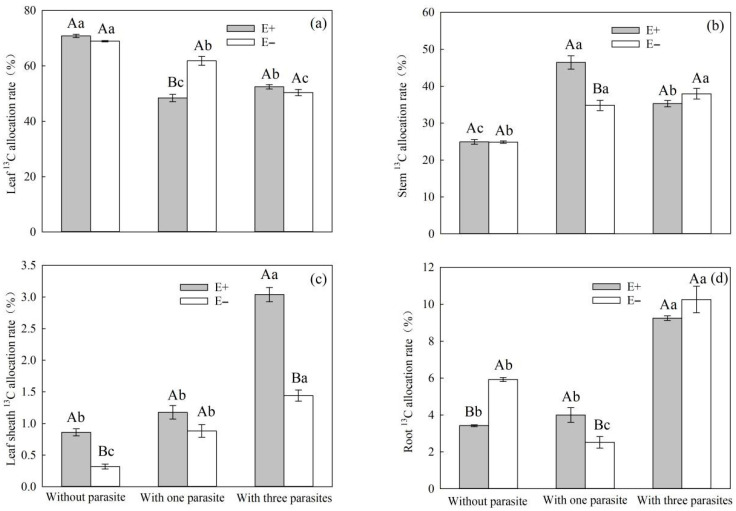
Effects of *P. kansuensis* density and endophyte status on the ^13^C allocation rate of the leaves, stems, leaf sheaths, and roots of *S. purpurea*. Note: Data are the mean standard errors. Different capital letters indicate significant differences between different *S. purpurea* endophyte statuses at the same *P. kansuensis* density (*p* < 0.05); Different lowercase letters indicate significant differences between different *P. kansuensis* densities in the same *S. purpurea* endophyte state (*p* < 0.05). One hemiparasite indicates that one *S. purpurea* plant was parasitized by one *P. kansuensis*, three hemiparasites indicate that one *S. purpurea* plant was parasitized by three *P. kansuensis*, and “without parasite” indicates *S. purpurea* growing alone; (**a**) Effects of *P. kansuensis* density and endophyte status on the ^13^C allocation rate of the leaves of *S. purpurea*; (**b**) Effects of *P. kansuensis* density and endophyte status on the ^13^C allocation rate of the stems of *S. purpurea*; (**c**) Effects of *P. kansuensis* density and endophyte status on the ^13^C allocation rate of the leaf sheaths of *S. purpurea*; (**d**) Effects of *P. kansuensis* density and endophyte status on the ^13^C allocation rate of roots of *S. purpurea*.

**Figure 11 microorganisms-11-02761-f011:**
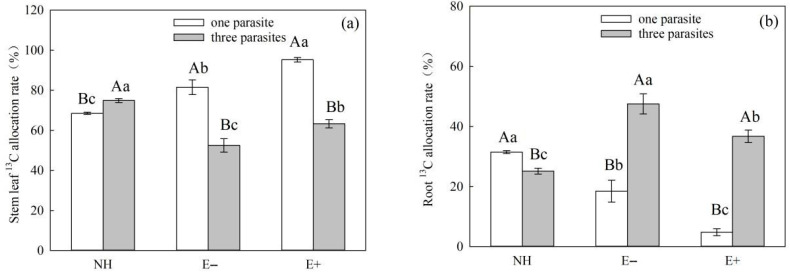
Effects of the *P. kansuensis* density and endophyte status of *S. purpurea* on the ^13^C allocation rate of the stem leaves, and roots of *P. kansuensis*. Note: Data are the mean standard errors. Different capital letters indicate significant differences between the different *P. kansuensis* densities in the same *S. purpurea* endophyte state (*p* < 0.05). Different lowercase letters indicate significant differences between the different *S. purpurea* endophyte states at the same *P. kansuensis* density (*p* < 0.05). NH indicates that *P. kansuensis* was not grown with *S. purpurea*; (**a**) Effects of *P. kansuensis* density and endophyte status on the ^13^C allocation rate of the stem leaves of *P. kansuensis*; (**b**) Effects of *P. kansuensis* density and endophyte status on the ^13^C allocation rate of the roots of *P. kansuensis*.

**Table 1 microorganisms-11-02761-t001:** Two-way ANOVA results for the effect of the endophyte fungal infection status (E) and parasitic density of the *P. kansuensis* (P) on the biomass (including the dry weight of the leaf sheath, stem, and leave), plant height, root length, and tiller number of *S. purpurea*.

Plant Growth Characteristics	Treatments	df	*F*	*p*
Dry weight of leaf	Endophyte fungal infection status (E)	1	1349.82	<0.01
	Parasitic density (P)	2	495.07	<0.01
	E × P	2	46.38	<0.01
Dry weight of stem	Endophyte fungal infection status (E)	1	583.64	<0.01
	Parasitic density (P)	2	754.09	<0.01
	E × P	2	110.38	<0.01
Dry weight of leaf sheath	Endophyte fungal infection status (E)	1	268.13	<0.01
	Parasitic density (P)	2	19.92	<0.01
	E × P	2	16.01	<0.01
Plant height	Endophyte fungal infection status (E)	1	13.16	<0.01
	Parasitic density (P)	2	17.85	<0.01
	E × P	2	0.61	0.56
Root length	Endophyte fungal infection status (E)	1	44.42	<0.01
	Parasitic density (P)	2	35.42	<0.01
	E × P	2	6.79	<0.05
Tiller number	Endophyte fungal infection status (E)	1	29.17	<0.01
	Parasitic density (P)	2	22.62	<0.01
	E × P	2	6.62	<0.05
Aboveground biomass	Endophyte fungal infection status (E)	1	2516.90	<0.01
	Parasitic density (P)	2	1060.58	<0.01
	E × P	2	136.99	<0.01
Underground biomass	Endophyte fungal infection status (E)	1	321.32	<0.01
	Parasitic density (P)	2	96.93	<0.01
	E × P	2	9.75	<0.01

**Table 2 microorganisms-11-02761-t002:** Two-way ANOVA results of the effects of the *P. kansuensis* parasitic density (P) and endophyte fungal infection status of *S. purpurea* (E) on the plant height, root length, aboveground biomass, and underground biomass of *P. kansuensis*.

Treatments	df	Plant Height		Root Length		Aboveground Biomass		Underground Biomass	
		*F*	*p*	*F*	*p*	*F*	*p*	*F*	*p*
Endophyte fungal infection status (E)	1	1.34	0.30	63.90	<0.01	82.18	<0.01	109.51	<0.01
Parasitic density (P)	2	0.87	0.35	2.59	0.13	423.85	<0.01	1220.92	<0.01
E × P	2	3.48	0.06	0.14	0.87	5.94	0.02	18.36	<0.01

**Table 3 microorganisms-11-02761-t003:** Two-way ANOVA results for the effect of the E endophyte fungal infection status (E) and parasitic density of the *P. kansuensis* (P) on the total carbon content of the leaves, stems, leaf sheaths, and roots of *S. purpurea*.

Treatments	df	Total Carbon Content of Leaves		Total Carbon Content of Stems		Total Carbon Content of Leaf Sheaths		Total Carbon Content of Roots	
		*F*	*p*	*F*	*p*	*F*	*p*	*F*	*p*
Endophyte fungal infection status (E)	1	1182.32	<0.01	600.50	<0.01	279.50	<0.01	254.72	<0.01
Parasitic density (P)	2	388.75	<0.01	716.82	<0.01	18.85	<0.01	50.27	<0.01
E × P	2	30.79	<0.01	107.20	<0.01	15.97	<0.01	5.06	0.025

**Table 4 microorganisms-11-02761-t004:** Two-way ANOVA results of the effects of the endophyte fungal infection status (E) and parasitic density of the *P. kansuensis* (P) on the total carbon content of the stems, leaves, and roots of the *P. kansuensis*.

Treatments	df	Total Carbon Content of Stems and Leaves		Total Carbon Content of Roots	
		*F*	*p*	*F*	*p*
Endophyte fungal infection status (E)	1	477.69	<0.01	1016.22	<0.01
Parasitic density (P)	2	81.78	<0.01	84.06	<0.01
E × P	2	6.04	0.015	21.20	<0.01

**Table 5 microorganisms-11-02761-t005:** Two-way ANOVA results of the effect of the endophyte fungal infection status (E) and parasitic density of the *P. kansuensis* (P) on the photosynthetic ^13^C content of the leaves, stems, leaf sheaths, and roots of *S. purpurea*.

Treatments	df	Leaf Photosynthetic ^13^C Content		Stem Photosynthetic ^13^C Content		Leaf Sheath Photosynthetic ^13^C Content		Root Photosynthetic ^13^C Content		Total Photosynthetic ^13^C Content	
		*F*	*p*	*F*	*p*	*F*	*p*	*F*	*p*	*F*	*p*
Endophyte fungal infection status (E)	1	267.43	<0.01	150.31	<0.01	495.90	<0.01	24.53	<0.01	444.19	<0.01
Parasitic density (P)	2	2182.23	<0.01	79.94	<0.01	54.22	<0.01	179.63	<0.01	1375.84	<0.01
E × P	2	14.85	0.01	18.84	<0.01	47.59	<0.01	89.94	<0.01	5.66	0.019

**Table 6 microorganisms-11-02761-t006:** Results of the two-way ANOVA showing the effects of the endophyte fungal infection status (E) and parasitic density of the *P. kansuensis* (P) on the ^13^C content of the stem leaves, roots, and total plant of the *P. kansuensis*.

Treatments	df	Stem Leaf ^13^C Content		Root ^13^C Content		Total ^13^C Content	
		*F*	*p*	*F*	*p*	*F*	*p*
Endophyte fungal infection status (E)	1	48.49	<0.01	202.27	<0.01	223.97	<0.01
Parasitic density (P)	1	105.40	<0.01	0.29	0.61	88.59	<0.01
E × P	1	12.08	<0.01	0.04	0.85	10.21	0.01

**Table 7 microorganisms-11-02761-t007:** Two-way ANOVA results for the effect of the endophyte fungal infection status (E) and parasitic density of the *P. kansuensis* (P) on the ^13^C allocation rate of the leaves, stems, leaf sheaths, and roots of *S. purpurea*.

Treatments	df	Leaf ^13^C Allocation Rate		Stem ^13^C Allocation Rate		Leaf Sheath ^13^C Allocation Rate		Root ^13^C Allocation Rate	
		*F*	*p*	*F*	*p*	*F*	*p*	*F*	*p*
Endophyte fungal infection status (E)	1	13.53	<0.01	9.67	<0.01	127.03	<0.01	5.11	0.04
Parasitic density (P)	2	172.25	<0.01	94.16	<0.01	188.24	<0.01	174.49	<0.01
E × P	2	35.68	<0.01	20.30	<0.01	30.85	<0.01	15.13	<0.01

**Table 8 microorganisms-11-02761-t008:** Results of the two-way ANOVA showing the effects of the endophyte fungal infection status (E) and parasitic density of the *P. kansuensis* (P) on the ^13^C allocation rate of the stem leaf and root of the *P. kansuensis*.

Treatments	df	Stem Leaf ^13^C Allocation Rate		Root ^13^C Allocation Rate	
		*F*	*p*	*F*	*p*
Endophyte fungal infection status (E)	1	94.36	<0.01	94.36	<0.01
Parasitic density (P)	2	14.51	<0.01	14.51	<0.01
E × P	2	43.11	<0.01	43.11	<0.01

## Data Availability

The original contributions presented in the study are included in the article. Further inquiries can be directed to the corresponding author.

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
