# Peer review of "Effect of Epichloë Endophyte on the Growth and Carbon Allocation of Its Host Plant Stipa purpurea under Hemiparasitic Root Stress"

_microorganisms, 2023, doi:10.3390/microorganisms11112761_

Round 1

Reviewer 1 Report

Comments and Suggestions for Authors

This manuscript describes the effect of fungal endophyte infection on the hemiparasitic interaction of the root parasitic plant Pedicularis kansuensis with the host plant Stipa purpurea. To analyse this effect, the authors performed experiments where they compared different growth parameters and C accumulation and allocation in a three component system: the host plant (S. purpurea), its beneficial Epichloe fungal endophyte and its root hemiparasitic plant P. kansuensis. Besides, the parasitation of the host plant by P. kansuensis was carried out by using two modalities: low density (one parasite per host plant) and high density (three parasites per host plant). The interaction of endophytes with parasitic plants has rarely been study, which makes this work original and interesting.

The three components system under study is complex, and the complexity is increased by the density factor included. Therefore, the authors should be very careful in explaining the results very clearly and without mistakes. However, this is not the case. The writing is quite confuse and contains huge mistakes that make very difficult to follow the text. The authors should extensively revise the text and correct all the errors before this manuscript can be considered to be accepted.

Other specific comments:

Abstract

The first time that the species names P. kansuensis and S. purpurea appear in the text should be written in the complete form. The same at the main text.

Introduction:

LINE 42. Reference 3 is not appropriate for the affirmation of endophytes obtaining nutrients from the host. Author should look for an appropriate one.

Line 43-46. Rewrite these sentences, they make no sense.

Line 52. Replace “hemiparasitic stress” by “hemiparasitic plants”

Lines 79-83. Rewrite this sentence, it makes no sense

Line 88. Which is the species of Epichloe detected in the sampled grasses?

Line 94. It should be “transferred to P. kansuensis” instead of “from P. kansuensis”. This mistake is recurrent throughout the entire text

Materials and methods

Line 104. Reference Bao et al. 39 should be at the end of this line

The authors should explain which is the origin of host and parasite seeds? One plant? Several different plants?

Line 113. I do not think that sand has been autoclaved at 120ºC for 48 hours

Lines 132-134. This explanation is not clear. If three plants are sealed and labeled with 13CO2, why are the remaining 3 plants labeled with “13C isotopes” and used as controls?

Line 143 Replace “schematic” by “scheme”

It is not clear when were tha samples taken. After 3 days of labelling?

Line 196. Materials and methods explain the calculation of “13C partition”, but the results and discussion speak about “allocation”. Please clarify.

Results

The authors should eliminate all qualitative reference to the significance, such as “exceedingly” or “extremely”. The level of significance is shown by the P value, it must not be adjectivized.

The authors should revise the entire text to correct the way to refer to “P. kansuensis parasitized S. purpurea” they recurrently use the wrong expression “P. kansuensis parasitic S. purpurea

The authors should revise the entire text and correct the mistakes in the numbers of the figures.

Lines 226-238. Eliminate these lines. They do not add to what is shown in the figure.

Lines 260-264. These affirmations are not true.

Lines 267-273, 307-312, 330-336, 357-367, 390-395, 413-423. Eliminate these lines. They do not add to what is shown in the figure and make the reading confusing.

Lines 290-292. The authors should mention that the result is not significant

Figures 4, 5, 6, 7, 8, 9, 10: describe the meaning of upper and lower case letters

Line 304. Eliminate “rapidly”

Line 374-378. This sentence should be better written

Line 402. See comment in material and methods about “13C partition” and “13C allocation”.

Line 437-446. Please revise. The affirmations do not fit to what is shown

Discussion

In general, the discussion is highly speculative. The author make affirmations that are not supported by results or are not explained in the correct order. Also, there are mistakes when describing the results and adding the numbers of the figures. As in the results section, the authors should revise the entire text to correct the way to refer to “P. kansuensis parasitized S. purpurea” they recurrently use the wrong expression “P. kansuensis parasitic S. purpurea”. In conclusion, the authors should deeply revise and improve the text.

Line 469. Eliminate “plant height”

Line 471. Replace “resistance” by “tolerance”

Line 474-478. Rewrite this sentence it does not make much sense. Why do the authors mention here plant morphology? They have not measured that trait.

Lines 479-490. The first sentences (lines 479 to 480) are contradictory with the second part of the paragraph. If the withering is due to the inhibition of nutrients obtaining by E+, why is the biomass higher? The withering could be to alkaloids produced by the endophyte. If there is an hypothesis about enhanced photosynthetic capacity and C accumulation, the authors should contrast it with their actual results

Lines 489-490. Eliminate this sentence, it is redundant.

Lines 491-493. Revise this sentence. It is not right.

Line 494. Replace demonstrate by suggest. There is no direct measurement of light capture. This should be related to the photosynthetic C results

Line 517. Eliminate “leaf sheaths”

Lines 538-543. Revise writing there are several mistakes in the use of prepositions. Besides, the affirmation is not true.

Lines 543-551. These sentences are high speculative and should be properly bases. To affirm an irregular distribution of the endophyte, the authors should measure its biomass in the plant.

Lines 560-563. This affirmation is not true. No significant results. Indicate the correct figure numbers

Line 570. Replace “from” with “to”. Indicate figure number

Lines 578-581. Explain this sentence better

Line 583. Add Figure 3

Conclusions.

What does “non-infected mean” E-?

Comments on the Quality of English Language

English is moderately good, although there are some confusions in the use of prepositions that should be carefully reviewed because make that text very confusing

Reviewer 2 Report

Comments and Suggestions for Authors

Manuscript: Effect of Epichloё Endophyte on the Growth and Carbon Allocation of its Host Plant Stipa purpurea Under Hemiparasitic Root Stress

 Even if the experimental approaches seem interesting, several experiments are not convincing and a lot of  information or details of experiments are missing. more attention should be given to the writing and data analyses.

Major Compulsory Revisions:

1)      In the methodology section is mentioned that the biological samples (seeds of S. purpurea and P. kansuensis) used in this research were collected/obtained from natural grassland. However the authors did not perform validation of the species plants used. Molecular analyses is nescessary to demostrate that the seeds/plants used belong to the specie purpurea and discard that these are of other species of the Stipa genus (x eg. Stipa capillacea, which is similar morphologically to S. purpurea). Similarly, is neccesary the validation of specie of the biological material used as P. kansuensis.

2)      the authors should show the evidence (x eg. Image) of the endophytic status of the seedlings used in their experiments.

3)      Table 1 is very confussing. While that in methodology is mentioned that this research was performed with 8 treatments, in the table 1 only are presented three treatments in relation with the 8 characteristics of plants evaluated. Of the 8 treatments realized, which of these are described in table 1? It is not clear.

4)      There is a lot troubles in relation to the data analyses and in consequence the results reported. For example: in Lines 223-225, the authors mentioned that plant height of  S. purpurea is affected by the parasitism. However, according to Figure 2 the plant height is not affected. Similarly, the root lenght only is affected in plants with less parasitism density. The text apper to suggest that the changes reported ocuurs in all treatments. This incongruence among those written in the manuscript and those showed in the graphics, figures and statistics analysis is present in various sections of the text. Is necessary minacious revision. There is a lot results reported which are not correct according to graphics and statics.

5)      The writing of the text make that the information to be very difficult follow. The authors make diverse assertions, particularly in  Results section, but without specific indicate to which of the treatments they are making reference. For example: Lines 230-233, is necessary indicate that the sentence is in relation only to the E+S.purpurea treatment. This trouble is throughout of the manuscript. Is necessary that the authors re-written for that the information to be clear and easy follow.

6)      Table 1: Include the data of the E- S. purpurea treatment. Likewise, consider necessary to modify the table 1 since that the way in that the information is presented make that it to be very difficult following. Make separation with lines or another form among the characteristics, and another modifications, could to help to better undestand of teh information presented.

7)      Figure 3 (footnote). It says: NH indicates that P. kansuensis was not grown with S. purpurea. However in the figure 3 is showed two bars into the NH column each one representing different density of parasitism (1 and 3, respectivelly). Then, what is really NH? It is not clear.

8)      In general, the manuscript need revision, re-written, and minuciosous data analyses. There is a lot of errors, mainly on reported results, that the authors should correct.

9)      Improve the quality of tables and figures.  The full words in tables should to be in the same lane.

Minor Essential Revisions:

The first once that are mentioned S. purpurea and P. kansuensis should to use full scientific name. Subsequently to use abbreviation.

There is a few errors in relation to literature cited in the manuscript. some are duplicate as reference of the same sentence or these aren´t according a the instructions of the journal

Foot note figure 2. It say: without parasite indicates P. kansuensis growing alone. Is it correct? Is not S. puprpea?

Foot note figure 3. Change Effects of P. kansuensis density and endophyte status on the growth of P. kansuensis by Effects of P. kansuensis density and endophyte status of S. purpurea on the growth of P. kansuensis

Comments on the Quality of English Language

Extensive editing of English language required

Round 2

Reviewer 1 Report

Comments and Suggestions for Authors

Thank you for the response and the changes made. However, not all the changes are satisfying. There are some incomplete issues.

I number the questions with the same numbers I did in my first report

Introduction:

question 2: This modification is still not clear and still makes no change. Please, revise

question 5.  The authors just check the endophytic status visually. They assume that it is E. gansuensis from former results, but they really do not check. They should warn about this in the text and take it into consideration, because it may be affecting to the variability of the results.

Materials and Methods

Question 1. Of course I understand that there are 2 plant species. However, my question was about the origin of the seeds of each species: a single plant or different plants. Particularly, this is important to know if there can be different endophytes affecting to the host plant S. purpurea

Results

Question 7. Replace “but there were no significant effects” by although the differences were not significant

Question 12. This sentence stil has no sense. Please, correct

Discussion.

Question 10. This sentence still is not correct. Differences are minimal and not significant. You cannot do that affirmation

Question 12. Replace “but there were no significant effects” by although the differences were not significant. Replace “while” by “however”

Comments on the Quality of English Language

Thank you for the response and the changes made. However, not all the changes are satisfying. There are some incomplete issues.

I number the questions with the same numbers I did in my first report

Introduction:

question 2: This modification is still not clear and still makes no change. Please, revise

question 5.  The authors just check the endophytic status visually. They assume that it is E. gansuensis from former results, but they really do not check. They should warn about this in the text and take it into consideration, because it may be affecting to the variability of the results.

Materials and Methods

Question 1. Of course I understand that there are 2 plant species. However, my question was about the origin of the seeds of each species: a single plant or different plants. Particularly, this is important to know if there can be different endophytes affecting to the host plant S. purpurea

Results

Question 7. Replace “but there were no significant effects” by although the differences were not significant

Question 12. This sentence stil has no sense. Please, correct

Discussion.

Question 10. This sentence still is not correct. Differences are minimal and not significant. You cannot do that affirmation

Question 12. Replace “but there were no significant effects” by although the differences were not significant. Replace “while” by “however”

Reviewer 2 Report

Comments and Suggestions for Authors

Manuscript: Effect of Epichloё Endophyte on the Growth and Carbon Allocation of its Host Plant Stipa purpurea Under Hemiparasitic Root Stress

 Even if, the authors performed some changes and modifications according to the first revision/observations, various of  the observations were not consider. This new version of the manuscript still need more work since that there some sentences confussing and incorrect in relation with the data, tables and graphics presented.

Major Compulsory Revisions:

1)      In the first revision, I pointed out: In the methodology section is mentioned that the biological samples (seeds of S. purpurea and P. kansuensis) used in this research were collected/obtained from natural grassland. However the authors did not perform validation of the species plants used. Molecular analyses is nescessary to demostrate that the seeds/plants used belong to the specie purpurea and discard that these are of other species of the Stipa genus (x eg. Stipa capillacea, which is similar morphologically to S. purpurea). Similarly, is neccesary the validation of specie of the biological material used as P. kansuensis.

 The justification made by authors for not carrying out the requested experimentation refers only to the experience in collection and in previous studies that they have carried out with the plant species of this study, however, this argument does NOT provide certainty or demonstrate that the species of plants used in this study correspond to those mentioned. There is still a doubt as to whether what they refer as S. purpurea is really this specie or that in reality it is another species of the Stipa genus with similar morphological characteristics to S. purpurea. The fact that there are reports that S. purpurea is the most abundant in the area where the seeds collection was carried out DOES NOT guarantee that it is the only species of this genus existing in that region. The authors should show some evidence that demonstrate that the biological material used corresond with the species plants mentioned (S. purpurea and P. kansuensis). On the other hand, I consider it appropriate to specify from how many plants  were obtained the seeds used in the study for the seddlings obtention and assays.

 2)      In the first revision, I made the follow observation: the authors should show the evidence (x eg. Image) of the endophytic status of the seedlings used in their experiments. Even if, the authors included a photograph of hyphae (figure 1), this image (according to the text lines 98-100) correspond only to the microscopic examination of the seeds collected from natural grassland which were used for the seedlings obtention. However, the observation that I made was in relation with seedlings used in the experiments. In particular, is necessary show image related with the text of lines 112-114. Show evidence that the treatments of the seeds (wash, desinfection, etc) didn´t interfer with the endophytic status mentiones (both, E+ and E- ).

3)      In Figure 1.: Is necessary that the authors describe the image. Hyphae? Fron who is this hyphae?. Tahe image is fron seed?. what is the arrow  indicanding?, etc. Is necessary that the figures have an complete description of the presented in the figures. Consider this latter in all the manuscript.

 4)      As I mentioned in first revision: in the methodology section is mentioned that this research was performed with 8 treatments; however according to the text of lines 117-128 this allow indentified only 6 treatments: 1) S.purpurea+E, 2) S. purpurea+E+1 P. kansuensis seedling, 3) S. purpurea+E+3 P. kansuensis seedling, 4)S.purpurea-E, 6) S. purpurea-E+1 P. kansuensis seedling,6) S. purpurea-E+3 P. kansuensis seedlings. Additionally, 1 treatment of P. kansuensis (line 125) is identified. Then,  which are the 8 treatments that you mentioned in the text?. In the table 1 only are presented three treatments in relation with the 8 characteristics of plants evaluated. Of the 8 treatments realized, which of these are described in table 1? It is not clear.

 The authors´s response  made in this revision is not clear.  In those that the authors refer as 8 treatments (line 128) is been including the control group? First, the authors made differences among control group and treatments group (lines 124-127) but when they refer to the 8 treatments apper to that in this phrase “8 treatments” included both control group and treatments group. It is necessary to clarify.

 5)      In the first revision I highlighted: There is a lot troubles in relation to the data analyses and in consequence the results reported. I pointed out that in various sections of the text there is incongruence among those written in the manuscript and those showed in the graphics, figures and statistics analysis.  For this, I recommended a minacious revision by the authors. Even if, in this new version the authors made changes in relation with that indicated, there is still a lot of errors amog the written and those presented in tables or graphics. For example: lines 212-214, the authors assert that in the system endophyte+parasitism there is significant effect in all evaluated characteristics, however tiller number and plant height showed values p>0.05. lines 223-226: wrong sentence. The graphs in Figure 3 show that plant height and root lenght do not undergo changes as stated in the sentence. Lines 226-229: This is incorrect. According to their graphs (figue 3), dry weight leaf sheath and plant height did not show significant changes in S. purpurea system without vs 1 plant of P. kansuensis and in E+S.purpurea without and with parasitism of 1 plant, respectively.

Lines 242-244. It is incorrect, according to table 2 the parasitic density had significant effect on both the aboveground biomass and  underground biomass of P. kan- suensis (Table 2, P < 0.01).

Lines 247-250: wrong phrase. there was no impactin Plant height, root length and  aboveground biomass while that underground biomass show significant changes in the system E+D.

 Is necessary that the authors make a minacious revision of ALL manuscript and re-written the text in such a way that don´t made incorrect phrases or incongruent statements in relation with the data.

 6)      Lines 220-223: Does this sentence refer to the system of S. purpurea+E + P. Kansuensis? In this case the sentence is erroneous since the plant height does not decrease in relation to the increase in the density of P. kansuensis. If the sentence refers to the S.purpurea + P. kansuensis system regardless of the infection status, the sentence is not correct either since both root length, plant heught and undergrond biomass did not experience changes in function of the density of P.kansuensi.

 7)      Lines 215-220: Were plants with 1 and 3 parasitic plants considered together? Was there no difference between the treatments (1 or 3 seedlings P. kansuensis) in the characteristics evaluated? specify this in the table, if the data presented in table 1 included both S. purpurea with 1 or 3 seedlings of parasitic plant or to which density of parasitism correspond the data of table 1.

 8)      How do the authors explain the fact that, based on this study, the endophyte X density system did not present statistically significant changes (p>0.05) in tiller number while that the figure 3 (a-h) do show changes in this parameter in the EXD (endophyte X density) system?

 9)      The experiment whose data are presented in table 2 this was not described or mentioned in methodology section.

 10)   In table 2, according to the text, the treatment labeled as Endophyte refer to E+S. purpurea, but if this is the case, how is that you evaluated the growth parameters of P. kansuensis?. I consider that this doubt is because to the little or no detailed description of the treatments evaluated, which should be made clear in the methodology section.

 11)   Lack of methodology related to data figure 5. It is necessary that authors present the methodology related to the data that they show.

 12)   Table 1: Include the data of the E- S. purpurea treatment. It would help to make visible the impact of the presence of Endophyte or parasitism on S. purpurea.

 Minor Essential Revisions:

a)      In Table 2. Change: Two-way ANOVA results of the effects of P. kansuensis density (D) and endophyte status (E) on the plant height,root length,aboveground biomass and underground biomass of P. kansuensis to Table 2. Two-way ANOVA results of the effects of P. kansuensis density (D) and endophyte status of S. purpurea  (E) on the plant height, root length, aboveground biomass and underground biomass of P. kansuensis. 

 b)      Eliminate point line 100 in (Figure 1). Change (Figure.1) to (Figure 1)

 c)       Add point line 146: were taken. The 13C labeled plant….

d)      In Table 3: separate each other the titles of the columns  (x eg. Total carbon content of…. )

 e)      Carefully review the text as there are minimal errors in the writing

Comments on the Quality of English Language

Minor editing of English language required

Author Response

Response to reviewer 2 comments
